# High-resolution structure and biochemical properties of the LH1–RC photocomplex from the model purple sulfur bacterium, *Allochromatium vinosum*

Kazutoshi Tani [1,2,12,13✉], Ryo Kanno[3,12], Ayaka Harada[4,12], Yuki Kobayashi[5], Akane Minamino[6], Shinji Takenaka[6], Natsuki Nakamura[5], Xuan-Cheng Ji[5], Endang R. Purba[7], Malgorzata Hall[7], Long-Jiang Yu [8], Michael T. Madigan[9], Akira Mizoguchi[2], Kenji Iwasaki [4], Bruno M. Humbel[10,11], Yukihiro Kimura [6,13✉] & Zheng-Yu Wang-Otomo [5,13✉]

The mesophilic purple sulfur phototrophic bacterium *Allochromatium* (*Alc.*) *vinosum* (bacterial family *Chromatiaceae*) has been a favored model for studies of bacterial photosynthesis and sulfur metabolism, and its core light-harvesting (LH1) complex has been a focus of numerous studies of photosynthetic light reactions. However, despite intense efforts, no high-resolution structure and thorough biochemical analysis of the *Alc. vinosum* LH1 complex have been reported. Here we present cryo-EM structures of the *Alc. vinosum* LH1 complex associated with reaction center (RC) at 2.24 Å resolution. The overall structure of the *Alc. vinosum* LH1 resembles that of its moderately thermophilic relative *Alc. tepidum* in that it contains multiple pigment-binding α- and β-polypeptides. Unexpectedly, however, six Ca ions were identified in the *Alc. vinosum* LH1 bound to certain α1/β1- or α1/β3-polypeptides through a different $Ca^{2+}$-binding motif from that seen in *Alc. tepidum* and other *Chromatiaceae* that contain $Ca^{2+}$-bound LH1 complexes. Two water molecules were identified as additional $Ca^{2+}$-coordinating ligands. Based on these results, we reexamined biochemical and spectroscopic properties of the *Alc. vinosum* LH1–RC. While modest but distinct effects of $Ca^{2+}$ were detected in the absorption spectrum of the *Alc. vinosum* LH1 complex, a marked decrease in thermostability of its LH1–RC complex was observed upon removal of $Ca^{2+}$. The presence of $Ca^{2+}$ in the photocomplex of *Alc. vinosum* suggests that $Ca^{2+}$-binding to LH1 complexes may be a common adaptation in species of *Chromatiaceae* for conferring spectral and thermal flexibility on this key component of their photosynthetic machinery.

A full list of author affiliations appears at the end of the paper.

The phototrophic bacterium *Allochromatium* (*Alc.*) *vinosum* is the most-investigated purple sulfur bacterium and has been widely used as a model for studies of anoxygenic photosynthesis and associated sulfur metabolism[1,2]. *Alc. vinosum* uses light energy to drive photophosphorylation and extract electrons from reduced sulfur compounds to produce the ATP and NADH needed for autotrophic growth. The photosynthetic apparatus of *Alc. vinosum* consists of two light-harvesting (LH) complexes, LH1 and LH2, and the reaction center (RC) complex. Both LH1 and LH2 contain multiple α- and β-polypeptides[3–6] with variable compositions[7,8], making it difficult for definitive structural analysis. A preliminary crystallographic study of the *Alc. vinosum* LH1–RC core complex was reported at 4.5–5.15 Å resolutions[9]; however, further efforts could not improve the resolution[10].

*Alc. vinosum* is a mesophilic species of the family *Chromatiaceae* and is phylogenetically related to the mild thermophile *Alc. tepidum*[11] and the thermophile *Thermochromatium* (*Tch.*) *tepidum*[12,13], with average nucleotide identities (ANI) of 90.8% and 85.7%[11], respectively. As a result, *Alc. vinosum* has been used as a reference for comparison of the thermostability of many isolated proteins with their counterparts from *Tch. tepidum* and *Alc. tepidum*, including RuBisCO[14,15], high-potential iron–sulfur protein (HiPIP)[16,17], cytochrome *c'* and flavocytochrome *c*[18], and the LH1–RC complex itself[19,20]. In addition to thermostability, spectroscopic properties of the photocomplexes from *Alc. vinosum* have been studied[21–26], and the organism's LH1–RC complex has been used as a reference point for the spectral changes observed in the $Ca^{2+}$-induced, redshifted LH1–RC complexes of other purple sulfur bacteria[20,27–30]. Surprisingly, however, despite extensive reference to *Alc. vinosum* photocomplexes in the literature, no high-resolution structures of these complexes have been reported.

Following structural determinations of the LH1–RC complexes from *Tch. tepidum*[31,32] and *Alc. tepidum*[33], here we report a high-resolution cryo-EM structure of the *Alc. vinosum* LH1–RC complex. A key discovery in our work is that metal ions unequivocally identified as $Ca^{2+}$ are bound in the LH1 complex. Based on this unexpected result and because the presence of $Ca^{2+}$ was neither suspected nor revealed in previous studies of the *Alc. vinosum* LH1–RC complex[11,20,33], we have reinvestigated this complex and determined the effects of $Ca^{2+}$ on its biochemical and spectroscopic properties. Moreover, we also report the effects of variations in illumination on the spectral characteristics and polypeptide composition of the purified *Alc. vinosum* LH1–RC complexes. After five decades of research on *Alc. vinosum*, our results provide a fresh new insight into the key photocomplex of this model purple sulfur bacterium and provide the necessary foundation for exploring the mechanisms by which it captures and converts solar energy.

## Results

**Structural overview of the *Alc. vinosum* LH1–RC.** The *Alc. vinosum* LH1–RC complexes used for cryo-EM analysis were purified using two different methods: (i) one-step extraction using *n*-dodecyl β-D-maltopyranoside (DDM) followed by sucrose density gradient centrifugation (hereafter designated as sucrose-density), and (ii) two-step solubilization by lauryldimethylamine *N*-oxide (LDAO) and *n*-octyl β-D-glucopyranoside (OG) followed by DEAE chromatography using DDM and $CaCl_2$ (see Methods for details). Both purification methods yielded essentially the same absorption spectra with an LH1-$Q_y$ maximum at 889 nm (Supplementary Fig. 1a); this differs from that (884 nm) reported for the *Alc. vinosum* LH1–RC purified using *n*-dodecyl phosphocholine (DDPC, an ionic detergent) and NaCl[20,28,29], and the reason for these differences in $Q_y$ maxima is described in the Discussion.

The cryo-EM structures of the *Alc. vinosum* LH1–RC complexes were determined at 2.24 Å resolution for both sucrose-density and Ca-DEAE purified samples (Fig. 1, Table 1, Supplementary Figs. 2–5). The two structures are virtually identical in both protein conformation and cofactor arrangement with a root-mean-square deviation (RMSD) of 0.37 Å for the mainchain Cα carbons (Supplementary Fig. 6b) but they differ in the number of divalent cations bound to the RC Cyt-subunit (Supplementary Fig. 7). Hereafter, we describe only the structure of the LH1–RC complex purified by the sucrose-density method because it was purified under physiological conditions that retained all natively bound Ca ions.

The overall structure of the *Alc. vinosum* LH1–RC resembles that of *Alc. tepidum* LH1–RC[33] with an RMSD of 1.62 Å for the mainchain Cα carbons (Fig. 1d) and shares some features with the LH1–RCs of *Tch. tepidum*[32] and the spectrally unusual *Thiorhodovibrio* (*Trv.*) strain 970[34] LH1–RCs. The *Alc. vinosum* LH1 is a closed ring structure composed of 16 pairs of αβ-polypeptides, 32 bacteriochlorophylls (BChl) *a*, and 16 all-*trans* spirilloxanthins that are uniformly distributed around the RC (Fig. 1a, Fig. 2a, Fig. 3a, Supplementary Fig. 6). Multiple forms of LH1 α- and β-polypeptides were confirmed in the high-resolution cryo-EM structure (see next section). The *Alc. vinosum* RC contains a tetraheme-cytochrome (Cyt) subunit (Fig. 1a) and four BChls *a*, two bacteriopheophytins *a*, one 15-*cis*-spirilloxanthin, a menaquinone (MQ)-8 at the $Q_A$ site and a ubiquinone (UQ)-8 at the $Q_B$ site (Fig. 3a). Light-induced $P^+/P$ absorption difference spectra showed that the reduced form of the *Alc. vinosum* RC special pair has an absorption maximum at 885 nm (Supplementary Fig. 8a).

Six metal ions bound to the *Alc. vinosum* LH1 α- and β-polypeptides were detected in the density map (Fig. 1b, c, Fig. 2, Supplementary Fig. 7a) and identified as $Ca^{2+}$ by inductively coupled plasma atomic emission spectroscopy (ICP-AES)[20] (Supplementary Fig. 9). Further quantitative measurements by ICP-AES placed the $Ca^{2+}$ stoichiometry at approximately 7 $Ca^{2+}$ per LH1–RC, which is comparable with the number observed from the cryo-EM density map (Supplementary Fig. 7) and very close to the actual number of 6 (see next two sections). The high-resolution structure of the *Alc. vinosum* LH1–RC also identified a total of 333 water molecules that are mainly distributed on the transmembrane surfaces and the surface of the membrane-extruded Cyt-subunit (Supplementary Fig. 6c).

**Arrangement of the multiple *Alc. vinosum* LH1 polypeptides.** Three pairs of *pufBA* genes (*pufB₁A₁*, *pufB₂A₂*, and *pufB₃A₃*) encoding LH1 β- and α-polypeptides are present in the *Alc. vinosum* genome[4–6]. Previous work detected two forms of α-polypeptides (α1 and α2) and two forms of β-polypeptides (β1 and β3) in the purified LH1 complex[7]. The cryo-EM structure of the *Alc. vinosum* LH1–RC complex identified three forms of α-polypeptides (six α1, nine α2, and one α3) and two forms of β-polypeptides (ten β1 and six β3) (Fig. 2a, b). Based on this result, we reexamined the LH1 polypeptide composition using reverse-phase HPLC analysis[33,35] and confirmed the presence of the single α3-polypeptide (Supplementary Fig. 10b). The α1-polypeptides of *Alc. vinosum* form face-to-face dimeric subunits specifically with β3-polypeptides through interactions in their N-terminal domains, whereas α2-polypeptides specifically pair with β1-polypeptides (Fig. 2b), a feature also observed in the *Alc. tepidum* LH1[33]. The single α3-polypeptide forms a subunit with the β1-polypeptide, likely due to its higher sequence similarity with the α2-polypeptide (Supplementary Fig. 11b). One of the α1-polypeptides and the α3-polypeptide interact extensively with the RC Cyt-subunit through their membrane-extruded C-terminal domains (Supplementary

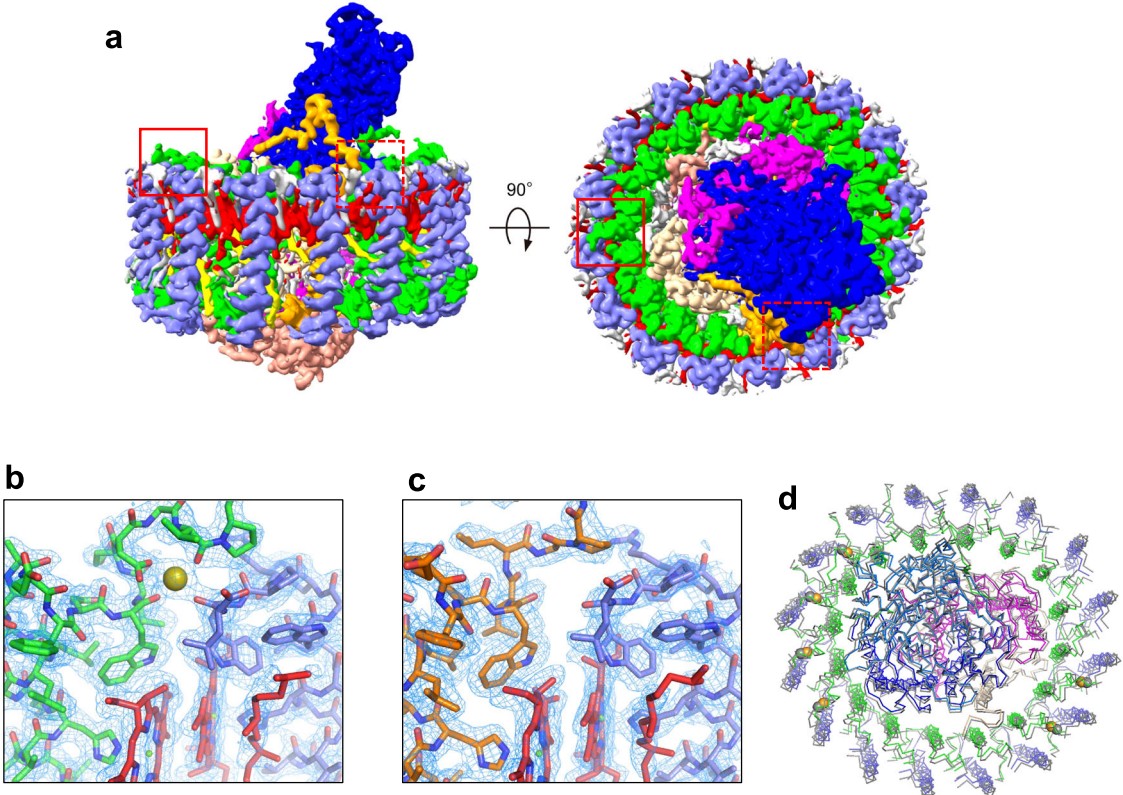

**Fig. 1 Overall structure and cofactor arrangement of the *Alc. vinosum* LH1–RC complex. a** Side and top views of surface representations of the LH1–RC parallel to the membrane and from the periplasmic side of the membrane, respectively. **b** A typical $Ca^{2+}$-binding site (marked in **a** by the red rectangle) with the density map around the C-termini of LH1 α1(green)- and β3(slate blue)-polypeptides. $Ca^{2+}$ bound to LH1 is shown by an olive sphere. **c** A typical $Ca^{2+}$-free site (marked in **a** by the red dashed rectangle) with the density map around the C-termini of LH1 α3(orange)- and β1(slate blue)-polypeptides. **d** Overlap view of the *Alc. vinosum* LH1–RC (colored) and that of *Alc. tepidum* (gray, PDB: 7VRJ) by superposition of Cα carbons of the RC-M subunits. Color scheme: LH1-α, green; LH1-β, slate blue; RC-L, wheat; RC-M, magenta; RC-H, salmon; RC-C, blue; BChls, red; carotenoids, yellow; lipids, gray.

Fig. 6d), strongly stabilizing the entire structure of the LH1–RC complex. Comparisons of the arrangement of *Alc. vinosum* LH1 αβ-polypeptides with those of *Alc. tepidum*, *Trv.* strain 970, and *Tch. tepidum* are shown in Supplementary Fig. 12.

Among the *Alc. vinosum* LH1 α-polypeptides, only α1-polypeptides are capable of binding $Ca^{2+}$ with β1- or β3-polypeptides (Fig. 2a, b), a feature also observed in the *Alc. tepidum* LH1[33]. Inspection of amino acid sequences revealed that the *Alc. vinosum* LH1 α1-polypeptide has a region of highly similar sequence to that of the $Ca^{2+}$-binding motif reported for the α-polypeptides in the $Ca^{2+}$-bound LH1 complexes (Fig. 2c) from *Tch. tepidum*[32], *Trv.* strain 970[34], and *Alc. tepidum*[33]. However, one $Ca^{2+}$-ligating residue (Ile) in the previously reported motifs is replaced by Val in the *Alc. vinosum* α1-polypeptide. By contrast, no such region is present in the *Alc. vinosum* α2- and α3-polypeptides (Fig. 2c). In addition, the shortest α2-polypeptides of *Alc. vinosum* exhibited a conformation in the C-terminus that would disrupt the $Ca^{2+}$-binding site and the longest α3-polypeptide adopted a different conformation from that of the α1-polypeptide around the $Ca^{2+}$-binding site, as is also the case in *Alc. tepidum*[33] (Supplementary Fig. 13). As a result, both α2- and α3-polypeptides of *Alc. vinosum* LH1 are unable to bind $Ca^{2+}$.

**The $Ca^{2+}$-binding sites and pigment arrangement.** A total of six Ca ions are located on the periplasmic side of the *Alc. vinosum* LH1 complex and are divided into two groups: four on one side of the LH1 ring and two on the opposite side (Fig. 3a). All $Ca^{2+}$ are positioned close to the LH1 BChl *a* molecules, and each $Ca^{2+}$ is

ligated by the mainchain oxygens of Trp44 and Val49, the side-chain of Asp47 in α1-polypeptides, and the mainchain oxygen of β3-Trp46 (or β1-Trp45) (Fig. 3b). Because of the high resolution of our structure, two water molecules were identified as additional $Ca^{2+}$ ligands (Fig. 3b); this leads to a hexa-coordinated octahedral structure, in which each $Ca^{2+}$ resides at the center of a plane formed by α1-Asp47, β3-Trp46 (or β1-Trp45) and oxygens of the two waters with the mainchain oxygen atoms of α1-Trp44 and α1-Val49 as axial ligands (Fig. 3b). In addition, several water molecules were identified near the $Ca^{2+}$-binding sites forming a hydrogen-bonding network with nearby residues and the $Ca^{2+}$-ligating waters (Fig. 3c) that stabilizes the $Ca^{2+}$-binding sites. The octahedral $Ca^{2+}$-binding pattern is similar to that of *Tch. tepidum*[32] but differs from that of *Alc. tepidum*[33]. In the latter case, both oxygen atoms in the carboxyl group of the α1-Asp47 are ligands to $Ca^{2+}$, whereas one of the carboxyl oxygens in the α-Asp (α1-Asp47 for *Alc. tepidum* and α-Asp49 for *Tch. tepidum*) forms a salt-bridge with a nearby guanidino group of β-Arg in the subunit (Fig. 3c).

Of the $Ca^{2+}$-ligating residues, sidechains of the α1-Trp44 and β3-Trp46 (or β1-Trp45) also form hydrogen bonds with the BChl *a* C3-acetyl group (Fig. 3d), implying that $Ca^{2+}$-binding may affect hydrogen-bonding for these residues. In fact, torsion angles of the C3-acetyl group tend to have smaller values for the $Ca^{2+}$-binding residues compared with those that do not bind $Ca^{2+}$ (Supplementary Fig. 14), despite relatively large deviations. The significance of this is that the torsion angle is thought to correlate with the $Q_y$ excitation energy[36,37], larger values resulting in more blue-shifted $Q_y$ peaks. Overall, the $Ca^{2+}$-binding effect on LH1 BChl *a* organization is likely limited because there are no

**Table 1 Cryo-EM data collection, refinement and validation statistics of the *Alc. vinosum* LH1–RC complexes.**

| | Sucrose-density purified LH1-RC complex (EMDB-37465, PDB-8WDU) | Ca²⁺-DEAE purified LH1-RC complex (EMDB-37466, PDB-8WDV) |
|---|---|---|
| *Data collection and processing* | | |
| Microscope | JEOL CRYO-ARM300II | TF Titan Krios |
| Camera | K3 | Falcon III |
| Magnification | 80,000 | 96,000 |
| Voltage (kV) | 300 | 300 |
| Electron exposure (e-/Å²) | 50 | 40 |
| Defocus range (μm) | −0.7 to −3.2 | −0.5 to −2.6 |
| Calibrated pixel size (Å) | 0.606 | 0.82 |
| Detector physical pixel size (μm) | 5 | 14 |
| Symmetry imposed | C1 | C1 |
| Initial particle images (no.) | 361,654 | 833,998 |
| Final particle images (no.) | 252,230 | 219,233 |
| Map resolution (Å) | 2.2 | 2.2 |
| FSC threshold | 0.143 | 0.143 |
| Map resolution range (Å) | 3.2–2.2 | 4.3–2.1 |
| *Refinement* | | |
| Initial model used (PDB code) | 7VRJ | 7VRJ |
| Model resolution (Å) | 2.3 | 2.3 |
| FSC threshold | 0.5 | 0.5 |
| Model resolution range (Å) | 135–2.2 | 135–2.2 |
| Map sharpening *B* factor (Å²) | –44 | –63 |
| *Model composition* | | |
| Non-hydrogen atoms | 26,274 | 26,261 |
| Protein residues | 2575 | 2585 |
| Ligands | 113 | 111 |
| Waters | 334 | 354 |
| *B factors (Å²)* | | |
| Protein | 27.4 | 45.9 |
| Ligand | 30.3 | 51.2 |
| Water | 23.6 | 47.8 |
| *R.m.s. deviations* | | |
| Bond lengths (Å) | 0.006 | 0.006 |
| Bond angles (°) | 2.821 | 2.796 |
| *Validation* | | |
| MolProbity score | 1.77 | 1.61 |
| Clashscore | 12.15 | 12.55 |
| Poor rotamers (%) | 1.63 | 0.98 |
| *Ramachandran plot* | | |
| Favored (%) | 97.95 | 98.2 |
| Allowed (%) | 2.05 | 1.8 |
| Disallowed (%) | 0 | 0 |

apparent differences in the Mg–Mg distances between BChls *a*, the length of the His–Mg(BChl *a*) coordination, and the length of the hydrogen bond for the Trp and BChl *a* C3-acetyl group between the Ca²⁺-bound and free LH1-αβ pairs. The latter was supported by the same wavenumber (1637 cm⁻¹) being observed for the C＝O stretching mode of the C3-acetyl groups in resonance Raman spectra (Supplementary Fig. 14c) used as a measure for estimating the strength of hydrogen bonding[30].

There were no apparent differences in average Mg–Mg and His–Mg distances in the *Alc. vinosum* LH1 complex from those of other purple bacteria (Supplementary Table 1). The long and short Mg–Mg distances and their difference well correlate with the *Alc. vinosum* LH1-$Q_y$ transition (889 nm, Fig. 4a), a correlation reported previously[30] showing that Mg–Mg distance differences larger than 0.8 Å tend to yield an LH1-$Q_y$ in the 870–890 nm range, whereas differences smaller than 0.8 Å (more homogeneous distances) typically yield an LH1-$Q_y$ beyond 900 nm for both BChl *a*- and BChl *b*-containing LH1 complexes.

**Biochemical and spectroscopic properties of the *Alc. vinosum* LH1–RC complex.** Because Ca²⁺-binding to select polypeptides in the *Alc. vinosum* LH1 complex was identified from the cryo-EM structure (Fig. 1, Supplementary Fig. 7), we examined the effects of Ca²⁺ on biochemical and spectroscopic properties of the complex. Removal of Ca²⁺ from the sucrose-density purified LH1–RC using EDTA resulted in a complex with an LH1-$Q_y$ at 886 nm, and subsequent addition of Ca²⁺ to the Ca²⁺-depleted LH1–RC restored its LH1-$Q_y$ to the original position (889 nm) at room temperature (Fig. 4a), revealing small but distinct differences in the absorption spectra of Ca²⁺-containing versus Ca²⁺-free complexes. Similar spectral differences were observed at 77 K (Supplementary Fig. 1b) and in room temperature spectra of the *Alc. vinosum* LH1–RC purified using NaCl instead of CaCl₂ in the DEAE chromatography step (Supplementary Fig. 1c); the latter indicates that Ca²⁺ can be replaced either partially or completely by Na⁺ during *Alc. vinosum* LH1–RC purification. These Ca²⁺ effects were more remarkable on thermostability of the *Alc. vinosum* LH1–RC complex. Incubation of the *Alc. vinosum* complex in the presence of Ca²⁺ at 60 °C—well above the maximum growth temperature of this species—for 80 min resulted in only a slight reduction of the LH1-$Q_y$ intensity, whereas similar treatment for the Ca²⁺-depleted LH1–RC led to loss of nearly half the absorbance (Fig. 4b), indicating that the Ca²⁺-bound *Alc. vinosum* LH1–RC is significantly more heat stable than the Ca²⁺-depleted complex.

Our previous study demonstrated that cultures of *Alc. vinosum* cells grew in the absence of Ca ions[29], and so here we examined the effects of Ca²⁺ on the spectroscopy of the isolated LH1–RC. The LH1–RC purified from cells of *Alc. vinosum* taken from a 7th serial subculture grown without Ca²⁺ supplementation showed an absorption maximum at 887 nm (Fig. 5a) that did not change upon addition of Ca²⁺. This indicates that polypeptide conformations around the Ca²⁺-binding sites in the LH1 complex of Ca²⁺-starved cells were irreversibly altered during growth, possibly as a mechanism to accommodate metal ions other than Ca²⁺.

We also investigated the effects of light intensity and quality on absorption spectra and polypeptide composition of the *Alc. vinosum* LH1 complex because different spectral forms and polypeptide compositions of the *Alc. vinosum* LH2 complex have been reported in cells grown under different illumination regimens[8]. *Alc. vinosum* intracytoplasmic membranes (ICM) contained slightly more LH2 than LH1 at low incandescent light intensity (LL, 10 μmol m⁻² s⁻¹) compared with that at middle light intensity (ML, 52 μmol m⁻² s⁻¹) (Supplementary Fig. 10a), and the two LH2 peaks at 804 nm and 848 nm showed similar intensities. By contrast, when cells were grown using an LED lamp (LED850, peak at 850 nm), although no apparent changes were evident in the proportion of LH1 expression in ICM compared with that at incandescent ML intensity, the two LH2 peaks (804 nm and 852 nm in this case) showed significantly different intensities, with a more intense 852-nm peak than 804-nm peak; this signals a marked change in the polypeptide composition of the *Alc. vinosum* LH2 between the two

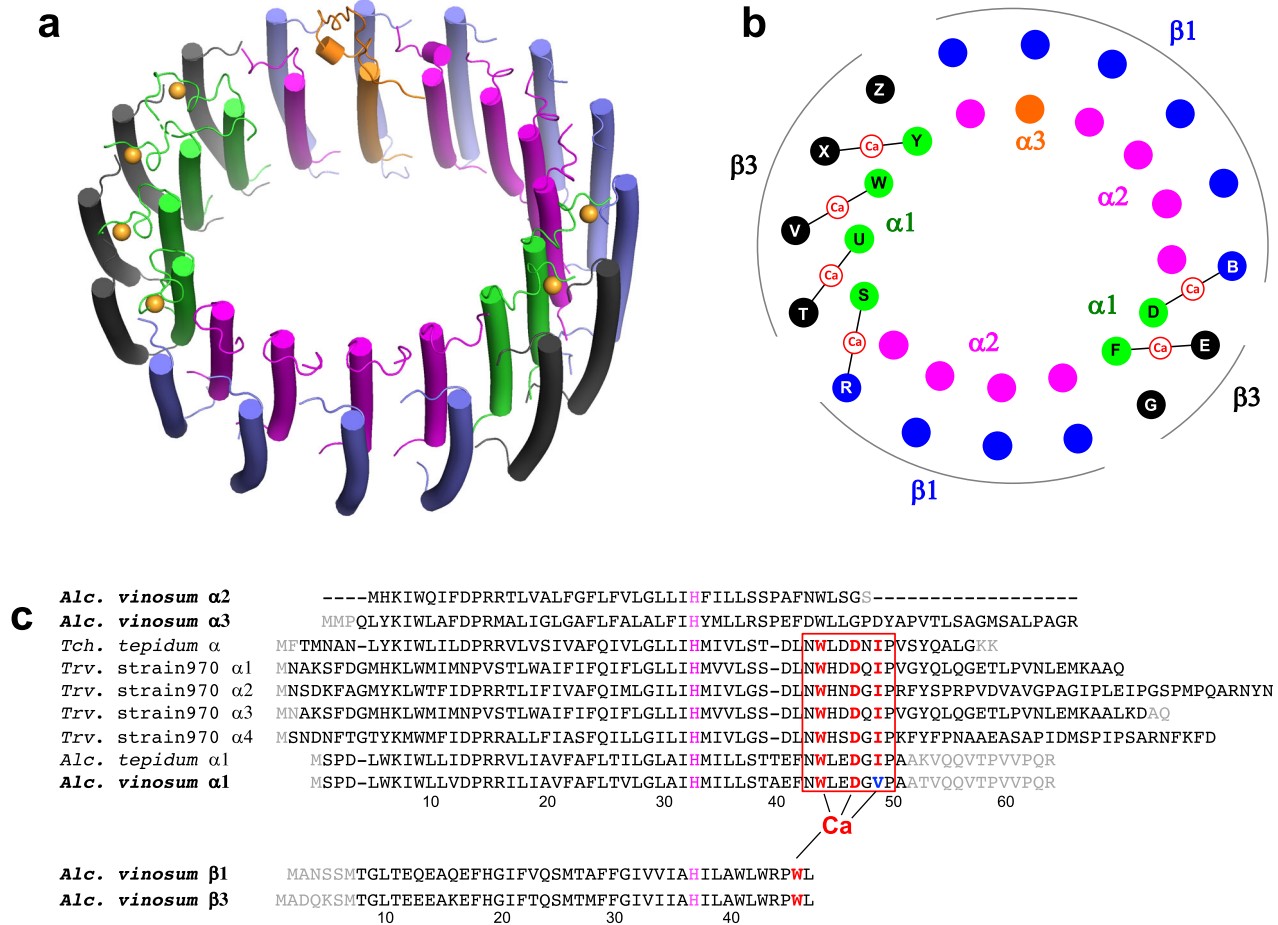

**Fig. 2 Arrangement of the *Alc. vinosum* LH1 multiple polypeptides and Ca$^{2+}$-binding motifs. a** Tilted view of the LH1–RC from the periplasmic side of the membrane. Color scheme: α1, green; α2, magenta; α3, orange; β1, blue; β3, black; Ca$^{2+}$, orange ball. **b** Illustration of the arrangement of the LH1 polypeptides. Letters in the colored circles denote chain IDs. Color scheme as in (**a**). **c** Sequence alignment of the *Alc. vinosum* LH1 polypeptides with those of *Tch. tepidum*, *Trv.* strain 970 and *Alc. tepidum* relative to the BChl *a*-coordinating histidine residues. Alignment of the α-polypeptides shows that the Ca$^{2+}$-binding motif WxxDxV is only present in the *Alc. vinosum* α1-polypeptides but is similar to the WxxDxI motif present in the Ca$^{2+}$-binding LH1 polypeptides from other purple sulfur bacteria.

illumination regimes. Despite these differences, the purified *Alc. vinosum* LH1–RC complexes exhibited essentially identical absorption spectra (Fig. 5b) and similar polypeptide compositions within the bounds of experimental errors (Supplementary Fig. 10b). These data suggest that changes induced by variations in light intensity and quality in *Alc. vinosum* mainly induce changes in the organism's abundant LH2 complexes rather than LH1–RC complexes.

## Discussion

The high-resolution structure of the *Alc. vinosum* LH1–RC complex revealed many details previously unsuspected, including in particular the presence of bound Ca$^{2+}$ and several bound water molecules. The overall structural features of the *Alc. vinosum* LH1–RC complex resemble those of the *Alc. tepidum* LH1–RC in both composition and arrangement of the multiple LH1 polypeptides (Fig. 1d, Fig. 2), likely due to their genetic similarity[11]. Nevertheless, Ca ions were unexpected in the *Alc. vinosum* LH1 complex but indeed were present and in the same number and sites as those for *Alc. tepidum* LH1[33]. The Ca$^{2+}$-binding motif in *Alc. vinosum* LH1 is composed of a region (WxxDxV) present only in the α1-polypeptides and a Trp residue in the C-terminal region of β1- or β3-polypeptides. Moreover, the WxxDxV Ca$^{2+}$-binding motif in *Alc. vinosum* differs from the WxxDxI motif in the LH1 α-

polypeptides of *Alc. tepidum*[33], *Tch. tepidum*[32] and *Trv.* strain 970[34] (Fig. 2c, Table 2), indicating that a substitution for Ile in the motif is not as critical for Ca$^{2+}$-binding as was previously thought[33]. Similar to the Ca$^{2+}$-ligating pattern in *Tch. tepidum* LH1[32], the Ca ions in *Alc. vinosum* LH1 are hexa-coordinated (including two water ligands) and form an octahedral structure, which differs from the pattern seen in *Alc. tepidum*[33]. Thus, the Ca$^{2+}$-binding motif in the LH1 complexes of purple sulfur bacteria can now be updated to a sequence of "WxxDx(I/V)".

Discovery of Ca$^{2+}$ in the *Alc. vinosum* LH1 prompted us to investigate the effects of Ca$^{2+}$ on biochemical and spectroscopic properties of its LH1–RC complex in more detail. Absorption maxima (Q$_y$) of certain purple bacterial LH complexes are sensitive to the detergents used for solubilization and/or treatment of the complexes[38-41], and the effects of detergents on Q$_y$ can be influenced by the presence of metal ions[41]. Generally, non-ionic detergents (such as DDM and OG) are milder and less affected by salts and tend to yield LH complexes with an identical or very close Q$_y$ to that observed in ICM. By contrast, ionic detergents (such as LDAO, DDPC, and sodium cholate) tend to be influenced by salts, resulting in LH complexes with different spectral forms and/or Q$_y$ from those in ICM. Detergent-induced spectral changes in a photocomplex signal an alteration of its tertiary structure that modifies pigment–pigment interactions[39].

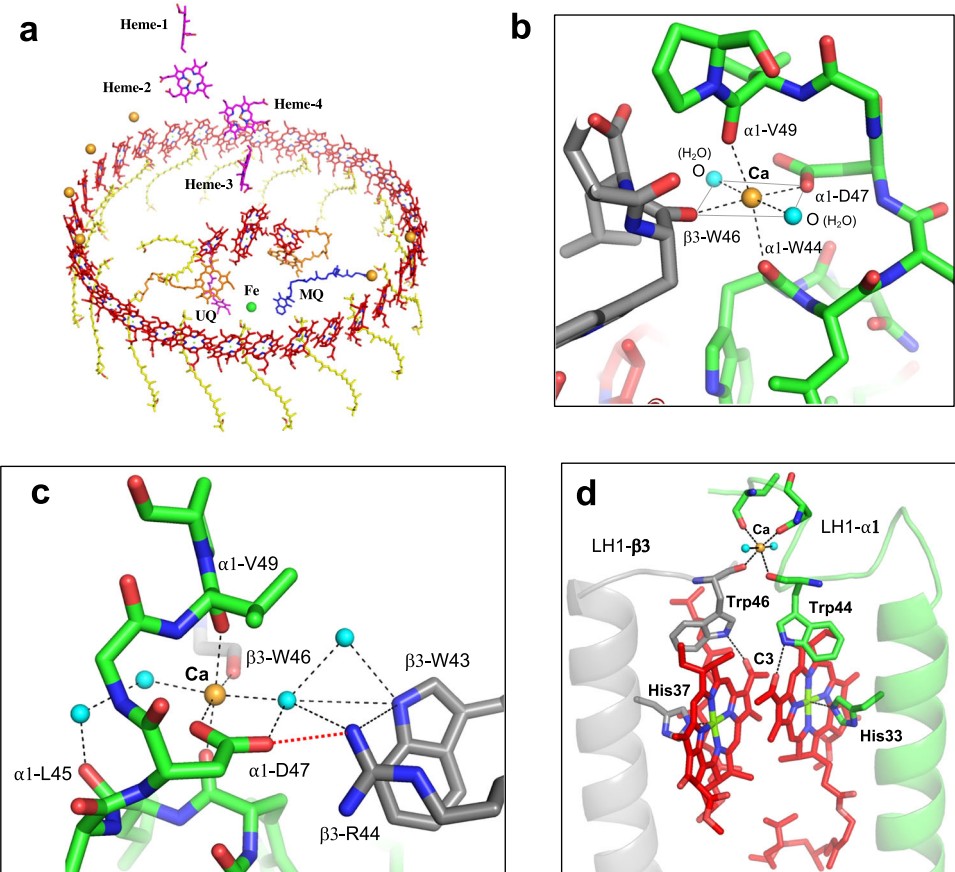

**Fig. 3 Ca²⁺- and BChl *a*-binding sites in the *Alc. vinosum* LH1 complex. a** Tilted view of the cofactor arrangement with the periplasm above and the cytoplasm below. **b** A typical Ca²⁺-binding site in the pair of α1/β3-polypeptides. **c** Hydrogen-bonding network (black dashed lines) formed by water molecules (cyan balls) around the Ca²⁺-binding site. A salt-bridge is between α1-Asp47 and β3-Arg44 (red dotted line). **d** A typical BChl *a*-binding site in an α1/β3 subunit showing that α1-Trp44 and β3-Trp46 coordinate to Ca²⁺ with their mainchain oxygens and form hydrogen bonds through their sidechains with the BChl *a* C3-acetyl groups.

However, the spectral changes caused by ionic detergents are often reversible upon removal or exchange of the detergents. Differing from the LH1 complexes of *Alc. tepidum*, *Tch. tepidum* and *Trv.* strain 970, the *Alc. vinosum* LH1 exhibited a detergent-sensitive $Q_y$ that was further affected by metal ions (Table 2, Supplementary Fig. 15). Because the Ca²⁺ effect on the *Alc. vinosum* LH1-$Q_y$ is small (2–3 nm) and similar to that (~3 nm) of the detergent, it had been overlooked in several experiments conducted previously using DDPC as a detergent, including the absorption spectrum, thermostability (Fig. 4) and a metal-exchanging FTIR measurement[42]. With this in mind, the subtle differences observed in the perfusion-induced FTIR difference spectra for the *Alc. vinosum* LH1–RC upon Sr²⁺/Ca²⁺ exchange[42] can now be considered significant changes. To confirm this, we further measured the Sr²⁺/Ca²⁺ (or Ca²⁺/Sr²⁺) FTIR difference spectra of the *Alc. vinosum* LH1–RC with increased acquisition numbers to improve the S/N ratio (Supplementary Fig. 8b). The results are compared with those reported for *Tch. tepidum* where, based on their isotopic shifts, the characteristic FTIR difference bands have been assigned to the vibrational modes for polypeptide backbones, amino acid side chains and BChl *a* surrounding the Ca ions[42,43]. Among these, two intensive differential bands at 1015/1007 cm⁻¹ and at 891/883 cm⁻¹ ascribed to the $\nu$ C=C and $\delta$ NCC$_m$ of BChl *a*, respectively, are useful marker bands for the conformational changes of the LH1 BChl *a*. In the *Alc. vinosum* LH1–RC FTIR spectra (Supplementary Fig. 8b), these differential bands were

largely reduced in intensity compared with those of *Tch. tepidum*, indicating that no significant conformational changes are induced in *Alc. vinosum* LH1 BChl *a* upon Ca²⁺-to-Sr²⁺ substitution. These results strongly support the conclusion that the *Alc. vinosum* LH1-$Q_y$ band should be, as was found to be the case, only slightly blue-shifted upon removal of Ca²⁺ (Fig. 4a).

Compared with the fully Ca²⁺-bound LH1 complexes of *Tch. tepidum* and *Trv.* strain 970, the partially Ca²⁺-bound LH1 complexes of *Alc. vinosum* and *Alc. tepidum* exhibit only small redshifts, and these may be attributed to the following factors. First, based on the results of resonance Raman spectroscopy, hydrogen-bonding interactions between BChl and LH1 polypeptides have been shown to correlate with the LH1-$Q_y$[30], and such interactions in the LH1 of *Alc. vinosum* and *Alc. tepidum* are much weaker than those in the *Tch. tepidum* and *Trv.* strain 970 LH1[20,28–30]. Second, the partial Ca²⁺-binding networks formed in the *Alc. vinosum* and *Alc. tepidum* LH1 are divided into two parts (Fig. 2b) and are thus much less extensive than those of the fully Ca²⁺-bound LH1 complexes from *Tch. tepidum* and *Trv.* strain 970. Ca ions function to "lock" the LH1 structure, and different network patterns formed in LH1 complexes have been shown to affect their $Q_y$ transitions[44]. Third, the BChl-coordinating C-terminal domains of α-polypeptides are either much shorter or more mobile in *Alc. vinosum* and *Alc. tepidum* than in *Tch. tepidum* and *Trv.* strain 970. This is supported by the fact that the C-terminal residues in the α1-polypeptides (6 copies) of *Alc. vinosum* and *Alc. tepidum* immediately following the Ca²⁺-binding sites are invisible in the cryo-EM density maps (gray

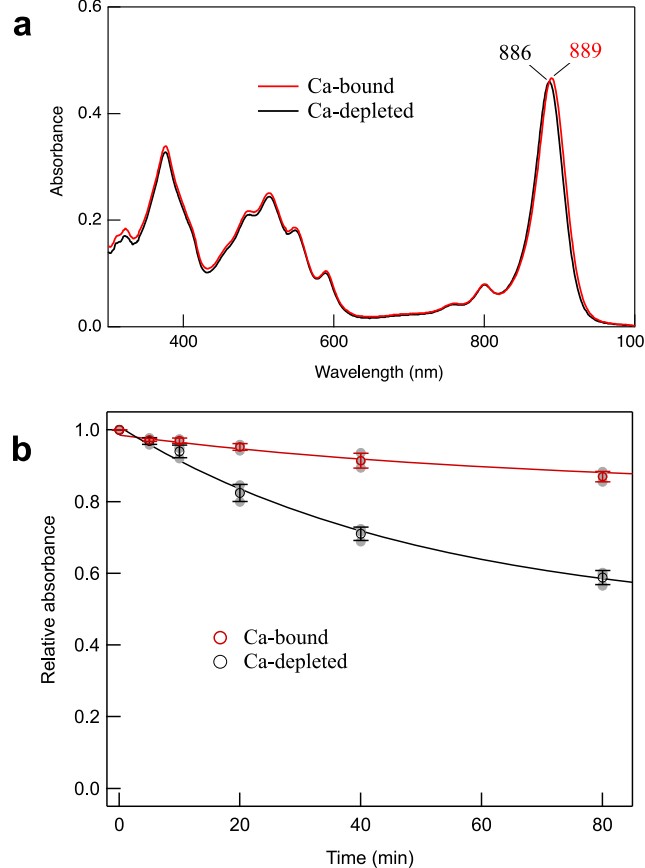

**Fig. 4 Absorption spectra and thermostability of the *Alc. vinosum* LH1–RC complex. a** Absorption spectra of the sucrose-density purified LH1–RC complex treated with 20 mM EDTA in 20 mM Tris-HCl (pH7.5) buffer containing 0.08% w/v DDM followed by removing the EDTA (Ca²⁺-depleted, black curve) and then adding 20 mM CaCl₂ (Ca²⁺-bound, red curve). **b** Time course of the relative LH1-$Q_y$ intensities upon incubation of the Ca²⁺-depleted and Ca²⁺-bound LH1–RC complexes at 60 °C. Gray points indicate raw data.

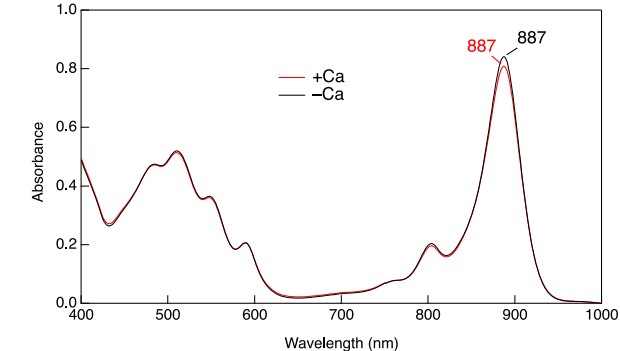

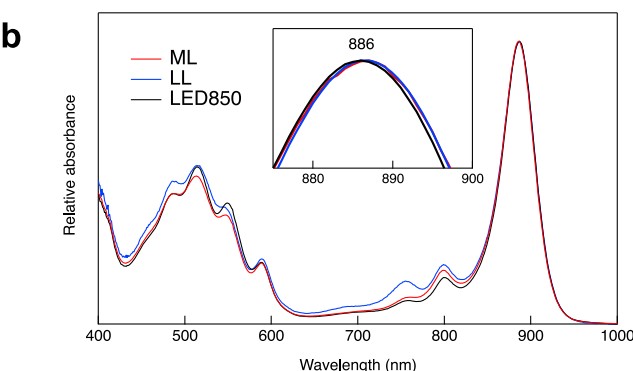

**Fig. 5 Effects of growth condition on the *Alc. vinosum* LH1–RC complex. a** Absorption spectra of the sucrose-density purified LH1–RC complexes from cells grown in Ca²⁺-free media (−Ca) followed by addition of 20 mM CaCl₂ (+Ca) in 20 mM Tris-HCl (pH7.5) buffer containing 0.08% w/v DDM. **b** Absorption spectra of the LH1–RC complexes purified using Na-DEAE chromatography from cells grown under incandescent illumination at middle light (ML) and low light (LL) intensities, or under LED illumination (LED850).

letters in Fig. 2c, Supplementary Fig. 16), and this indicates that large fluctuations and/or disorders are present that could destabilize Ca²⁺-binding sites. Such structural fluctuations and static disorders influence LH1-$Q_y$ and cause an inhomogeneous broadening of the absorption band[45–47], a result also observed with the *Alc. vinosum* LH1 where a large $Q_y$ redshift and band narrowing occurred at 77 K (Supplementary Fig. 1b). Fourth, it has been shown that point charge and protein polarity can induce a large $Q_y$ redshift of the BChl in a hydrophobic environment[30,36,48,49]. And finally, the number of Ca ions present in an LH1 and their close proximity to BChl *a* (Fig. 3d) also contributes to the differences observed in the LH1-$Q_y$ of partial and fully Ca²⁺-bound complexes. It is unlikely that any of the above factors affect LH1 absorbance independently but instead function in combination to confer the spectral changes observed in the *Alc. vinosum* LH1.

Despite Ca²⁺-incorporation into LH1 when Ca²⁺ is available, *Alc. vinosum* cells are also capable of growing phototrophically in a Ca²⁺-free medium[29] with repeated serial subcultures without a noticeable effect on either growth rate or growth yield. This is in sharp contrast to *Tch. tepidum*, *Trv.* strain 970 and *Alc. tepidum*, all of which exhibited a strong Ca²⁺-dependence for phototrophic growth[20,29]. Beside Ca²⁺, *Tch. tepidum* cells can grow when Ca²⁺ is replaced by Sr²⁺, but no other divalent cations support growth in the absence of Ca²⁺[50]. The LH1 of Sr²⁺-grown *Tch. tepidum* shows a $Q_y$ at 888 nm that shifts to 908 nm (rather than to the 915 nm of wild-type) by addition of Ca²⁺, implying irreversible changes in the protein conformation around Ca²⁺-binding pockets during Sr²⁺-dependent growth[50]. A subsequent study[51] on Sr²⁺-grown *Tch. tepidum* revealed preferential truncations of C-terminal residues in the LH1 α-polypeptides that play important roles in stabilizing the LH1 structure and modifying its spectral properties. Such polypeptide modifications may also explain the irreversible feature observed for LH1-$Q_y$ from *Alc. vinosum* cells grown in the absence of Ca²⁺ (Fig. 5a).

To date, Ca²⁺ has been identified in the LH1 complexes of all purple sulfur phototrophic bacteria with known LH1 structures. Thus, it is possible that Ca²⁺-binding to LH1 is a universal feature of *Chromatiaceae*, the variable Ca²⁺ amounts and their topology in the complex reflecting a flexible strategy for inhabiting ecological niches that differ in temperature, light availability and quality, or levels of nutritional resources.

## Methods

**Cultivation of *Alc. vinosum* cells.** *Alc. vinosum* strain D (DSM 180 ᵀ) was used in this study. Unless otherwise stated, cells of *Alc. vinosum* were cultivated phototrophically (anoxic/light) in a mineral medium[11] containing 0.34 mM CaCl₂ at room temperature for 7 days under incandescent illumination at middle-light intensity (ML, 52 μmol m⁻² s⁻¹). For light experiments, *Alc. vinosum* cultures were also cultivated under low-light intensity incandescent light (LL, 10 μmol m⁻² s⁻¹) or with an LED panel (ISL-150 × 150II85, CCS Inc., Japan) having peak wavelength emission at 850 nm (LED850). To investigate Ca²⁺ effects, *Alc.*

**Table 2 Characteristics of Ca²⁺-binding and its effects on the LH1 complexes of different purple sulfur bacteria.**

|  | Number of Ca²⁺ | Ca²⁺-binding motif | αβ-polypeptide | LH1-$Q_y$ (nm) | Ca²⁺ effect on LH1-$Q_y$ | Ca²⁺ effect on stability |
|---|---|---|---|---|---|---|
| *Alc. vinosum* | 6 (partial) | WxxDxV | Multiple | 889 | 2–3 nm | Moderate |
| *Alc. tepidum* | 6 (partial) | WxxDxI | Multiple | 890 | 8 nm | Moderate |
| *Tch. tepidum* | 16 (full) | WxxDxI | Single | 915 | 35 nm | Significant |
| *Trv.* strain 970 | 16 (full) | WxxDxI | Multiple | 959 | 84 nm | Significant |

*vinosum* cells were cultivated in a Ca²⁺-free mineral medium. The LED850-illuminated and Ca²⁺-free cells were harvested after 6 and 7 subcultures, respectively.

**Preparation of LH1–RC complexes**. The *Alc. vinosum* LH1–RC complexes for cryo-EM analysis were purified using two different methods. In the first method, LH1–RC complexes were prepared by one-step extraction from ICM using 1.0% w/v DDM, followed by sucrose density gradient centrifugation (five-stepwise concentrations: 10, 17.5, 25, 32.5, and 40% w/v) in 20 mM Tris-HCl (pH 8.0) buffer containing 0.05% w/v DDM. The LH1–RC band was collected and concentrated for cryo-EM measurements. In the second method, ICM were first treated with 0.35% LDAO to remove excess LH2, followed by solubilization using 1.0% w/v OG. Then, crude LH1–RC complexes were loaded onto a DEAE column equilibrated with 20 mM Tris-HCl buffer (pH 7.5) containing 0.1% w/v DDM. The LH1–RC fractions were eluted by a linear gradient of $CaCl_2$ from 0 mM to 100 mM and collected for cryo-EM analysis. Both purification methods yielded essentially the same absorption spectra with the LH1-$Q_y$ maxima at 889 nm (Supplementary Fig. 1a). The major difference between the two isolation methods is that no Ca²⁺ or other salts were used in the sucrose-density purification. LH1–RC complexes from *Alc. vinosum* cells grown in a Ca²⁺-free medium were purified by sucrose-density. LH1–RC complexes from cells grown at different light intensities and LED850 were purified by DEAE chromatography with a linear gradient (0 mM to 200 mM) of NaCl.

**Cryo-EM data collection**. Proteins for cryo-EM were concentrated to 5.2 and 4.0 mg/ml of the sucrose-density and Ca-DEAE purified samples, respectively. Three microliters of the protein solution were applied on glow-discharged holey carbon grids (200 mesh Quantifoil R1.2/1.3 and R2/2 molybdenum) that had been treated in a PIB-20 (Shinku Device) and Solarus-2 (Gatan) for 90 s and 30 s for the sucrose-density and Ca-DEAE purified samples, respectively, and then plunged into liquid ethane at −182 °C using an EM GP2 plunger (Leica, Microsystems, Vienna, Austria). The applied parameters were a blotting time of 6 s and humidity of 90% at 4 °C. Data from the sucrose-density sample were collected on a CRYO-ARM300II (JEOL) electron microscope at 300 kV equipped with a K3 camera (Gatan). An in-column energy filter with a slit width of 20 eV was inserted for acquisition of movie frames. Data from the Ca-DEAE sample were collected on a Titan Krios (Thermo Fisher Scientific) electron microscope at 300 kV equipped with a Falcon-III camera (Thermo Fisher Scientific). Movies were recorded using SerialEM[52] and EPU at a nominal magnification of 80 K and 96 K in CDS and counting mode and a pixel size of 0.606 and 0.820 Å at the specimen level for the sucrose-density and Ca-DEAE purified samples, respectively. The dose rate was 25.0 and 1.32 e- per Å² per second at the specimen level, and the exposure time was 2.0 and 30.15 s, resulting in an accumulated dose of 50.0 and 40.0 e− per Å² for the sucrose-density and Ca-DEAE purified samples, respectively. Each movie includes 50 and 40 fractioned frames for the sucrose-density and Ca-DEAE purified samples, respectively.

**Image processing**. All of the stacked frames were subjected to motion correction with MotionCor2[53], and defocus was estimated using CTFFIND4[54]. A total of 361,654 and 833,998 particles were selected from 7704 and 3114 micrographs for the sucrose-density and Ca-DEAE purified samples, respectively, using crYOLO[55] and RELION3.1[56]. All of the picked particles were further analyzed with RELION, and 359,604 and 338,772 particles were selected by 2-D classification and 1st 3D classification for the sucrose-density and Ca-DEAE purified samples, respectively. They were divided into four classes by 3-D classification resulting in only one good class containing 252,230 and 219,233 particles for the sucrose-density and Ca-DEAE purified samples, respectively. The initial 3-D model was generated in RELION. The 3-D auto refinement without any imposed symmetry (C1) produced two maps at 2.24 Å resolution after contrast transfer function refinement, Bayesian polishing, masking, and post-processing. These particle projections were then subjected to subtraction of the detergent micelle density followed by 3-D auto refinement to yield final maps with a resolution of 2.24 Å according to the gold-standard Fourier shell correlation using a criterion of 0.143. The local resolution maps were calculated on RELION.

**Model building and refinement of the LH1–RC complex**. The atomic model of the *Alc. tepidum* LH1–RC (PDB code 7VRJ) was fitted to the cryo-EM map obtained for the *Alc. vinosum* LH1-RC using Chimera[57]. Amino acid substitutions and real space refinement for the peptides and cofactors were performed using COOT[58]. The C-terminal regions of the LH1 α3-subunit were modelled ab-initio based on their density. The manually modified model was real-space-refined on PHENIX[59], and the COOT/PHENIX refinement was iterated until the refinements converged. Finally, the statistics calculated using MolProbity[60] were checked. Figures were drawn with the Pymol Molecular Graphic System (Schrödinger)[61], UCSF Chimera[57] and Chimera-X[62].

**Biochemical analyses of the *Alc. vinosum* LH1–RC complex**. Metal ion identification in the *Alc. vinosum* LH1–RC complex purified by sucrose-density was carried out on an ICP-AES spectrophotometer (ICPS-7510, Shimadzu). Qualitative measurements detected three metal ions (Ca, Mg and Fe) above 0.1 ppm in the LH1–RC at a concentration of $A_{889} = 1.3\,cm^{-1}$. Further quantitative measurements on the same LH1–RC sample were conducted using Multielement Standard Solution W-II (Wako Pure Chemical Industries, Ltd.) that contains 1000 mg/L for each Ca, Fe, and Mg in 1 M $HNO_3$ solution and was diluted to appropriate concentrations used for calibration. Detection wavelengths were 393.366 nm for Ca, 279.553 nm for Mg and 259.940 nm for Fe. The *Alc. vinosum* LH1–RC complex purified by sucrose-density was treated by 20 mM EDTA in 20 mM Tris-HCl (pH 7.5) buffer containing 0.08% w/v DDM on ice, followed by removing EDTA to yield Ca-depleted LH1–RC (Fig. 4a), and then adding 20 mM $CaCl_2$ to yield Ca-bound LH1–RC (Fig. 4a). Thermal degradations of the Ca-bound and Ca-depleted LH1–RCs were monitored via the LH1-$Q_y$ intensity after incubation at 60 °C for 0–80 min (Fig. 4b). The LH1–RC complex purified by sucrose-density from cells grown in a Ca²⁺-free

medium was treated with 20 mM $CaCl_2$ in 20 mM Tris-HCl (pH 7.5) buffer containing 0.08% w/v DDM.

**Statistics and reproducibility.** Thermal stability measurements (Fig. 4b) were performed in triplicate (three individual assays) independently to verify reproducibility. Experimental errors for all absorption spectra were within 1 nm.

**Reporting summary.** Further information on research design is available in the Nature Portfolio Reporting Summary linked to this article.

## Data availability

Maps and models have been deposited in the EMDB and PDB with the accession codes: EMD-37465 and PDB-8WDU for the *Alc. vinosum* LH1–RC purified by sucrose-density; EMD-37466 and PDB-8WDV for the *Alc. vinosum* LH1–RC purified by Ca-DEAE. The numerical source values underlying Fig. 4 can be found in Supplementary Data 1. All other data are available from the authors upon reasonable request.

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

## Acknowledgements
This research was partially supported by Platform Project for Supporting Drug Discovery and Life Science Research (Basis for Supporting Innovative Drug Discovery and Life Science Research (BINDS)) from AMED under Grant Numbers JP21am0101118 and JP21am0101116, and JP23ama121004. R.K., E.R.P., M.H., and B.M.H. acknowledge the generous support of the Okinawa Institute of Science and Technology (OIST), Scientific Computing & Data Analysis Section and Scientific Imaging Section at OIST and the Japanese Cabinet Office. R.K. acknowledges the support from Prof. Tsumoru Shintake. L.-J.Y. acknowledges support of the National Key R&D Program of China (Nos. 2021YFA0909600 and 2019YFA0904600). M.T.M. was supported in part by NASA Cooperative Agreement 80NSSC21M0355. This work was supported in part by JSPS KAKENHI Grant Numbers 20H05086, 20H02856, and 22K06111.

## Author contributions
Z.-Y.W.-O., Y.Ki., and K.T. designed the work. K.T., R.K., A.H., Y.Ko., A.Min., N.N., X.-C.J., E.R.P. and M.H. performed the experiments, K.T., R.K., S.T., L-J.Y., M.T.M., A.Miz., K.I., B.M.H. Y.Ki. and Z.-Y.W.-O. analyzed data, Z.-Y.W.-O., K.T., Y.Ki. and M.T.M. wrote the paper.

## Competing interests
The authors declare no competing interests.

## Additional information

[1]Center for Computational Sciences, University of Tsukuba, 1-1-1 Tennodai, Tsukuba, Ibaraki 305-8577, Japan. [2]Graduate School of Medicine, Mie University, 1577 Kurimamachiyacho, Tsu 514-8507, Japan. [3]Quantum Wave Microscopy Unit, Okinawa Institute of Science and Technology Graduate University (OIST), 1919-1, Tancha, Onna-son, Kunigami-gun, Okinawa 904-0495, Japan. [4]Life Science Center for Survival Dynamics, Tsukuba Advanced Research Alliance (TARA), University of Tsukuba, 1-1-1 Tennodai, Tsukuba, Ibaraki 305-8577, Japan. [5]Faculty of Science, Ibaraki University, Mito 310-8512, Japan. [6]Department of Agrobioscience, Graduate School of Agriculture, Kobe University, Nada, Kobe 657-8501, Japan. [7]Scientific Imaging Section, Research Support Division, Okinawa Institute of Science and Technology Graduate University (OIST), 1919-1, Tancha, Onna-son, Kunigami-gun, Okinawa 904-0495, Japan. [8]Photosynthesis Research Center, Key Laboratory of Photobiology, Institute of Botany, Chinese Academy of Sciences, Beijing 100093, China. [9]School of Biological Sciences, Program in Microbiology, Southern Illinois University, Carbondale, IL 62901, USA. [10]Provost Office, Okinawa Institute of Science and Technology Graduate University (OIST), 1919-1, Tancha, Onna-son, Kunigami-gun, Okinawa 904-0495, Japan. [11]Department of Cell Biology and Neuroscience, Juntendo University, Graduate School of Medicine, Tokyo 113-8421, Japan. [12]These authors contributed equally: Kazutoshi Tani, Ryo Kanno, Ayaka Harada. [13]These authors jointly supervised this work: Kazutoshi Tani, Yukihiro Kimura, Zheng-Yu Wang-Otomo. ✉email: ktani@ccs.tsukuba.ac.jp; ykimura@people.kobe-u.ac.jp; wang@ml.ibaraki.ac.jp

