## [Peer Review File · Communications Biology]

Reviewers' comments:

Reviewer #1 (Remarks to the Author):

Tani et al. describe a structural and biochemical characterization of the LHI-RC complex from *Allochrochromatium vinosum*. The complex contains various unexpected Ca ions bound to certain alpha and beta subunits of LHI, which suggests that Ca ion binding sites may be a common feature in Chromatiaceae. The role of these Ca ions are determined which especially influence the thermostability of the LHI-RC complexes.

In general, the structure is quite similar to that found in *Allochrochromatium tepidum*, which slightly diminishes the significance. However, the approaches to determine the roles of the Ca ions, the interpretation of the results, and the communication of that information in the context of existing literature are excellent. The manuscript therefore helps the field to better understand these well studied complexes in a new light.

Especially noteworthy are the following:

- (a) the high resolution achieved that allows for reliable structural determination (e.g., differentiating LHI alpha and beta subunits) and subsequent comparison with other structures,
- (b) the use of ICP-AES to determine the identity of metal ions in the structure, and
- (c) the comparison and biochemical characterization of the complexes with and without calcium

Thus, I enthusiastically recommend the manuscript for publication. My only concern is the use of terms like "best-studied" and "best-understood", especially in the title. These are somewhat subjective, so I suggest removing them.

Reviewer #2 (Remarks to the Author):

Review of the manuscript entitled: "High-resolution structure and biochemical properties of the LH1-RC photocomplex from the best-studied purple sulfur bacterium, *Allochrochromatium vinosum*" by Tani et al.

This manuscript is exciting as it reports the cryo-EM structure of LH1-RC isolated from the mesophilic purple sulfur photosynthetic bacterium *Alc. Vinosum*. This bacterium has been actively studied as a model organism for non-oxygenic photosynthesis, but the resolution of the structure of LH1 has been limited to a 10 Å resolution. In this paper, the authors report a high-resolution cryo-EM structure of LH1-RC isolated from *Alc Vinosum*. The findings by Tani et al. fill this knowledge gap and will be helpful to photosynthesis researchers. The authors found that the overall structure of *Alc. Vinosum* was very similar to that of *Alc. Tepidum*, but there were differences in the Ca binding site and mode of binding. The authors investigated the relationship between Ca and thermal stability by biochemical and spectroscopic experiments. The results indicate that Ca binding is a common factor for the LH1 complex to confer thermal stability to Chromatiaceae species. This may be of interest to structural biologists in general, as it may be one of the strategies by which protein complexes acquire thermal stability and adapt to their environment. Considering the specialized topics of the journal *Communications Biology* and its broad readers, this paper is worthy of publication. However, the reviewer feels that the following points should be addressed before accepting the manuscript.

The LH1-RC supercomplex from *Alc. Vinosum* contains multiple polypeptides of $\alpha 1$, $\alpha 2$, $\alpha 3$ and

polypeptides of $\beta 1$ and $\beta 2$. The composition of these polypeptides in the supercomplex and the basis of supercomplex formation are revealed in this manuscript. Namely, $\alpha 1$ - binds specifically to $\beta 3$, and $\alpha 2$ specifically to $\beta 1$, which is also seen in the *Alc. Tepidum* LH1-RC structure. The structure also shows that only $\alpha 1$ binds Ca with $\beta 1$ and $\beta 3$. As a result, six Ca-binding sites are present in the supercomplex. Furthermore, the authors estimated the number of Ca, Mg, and Fe in the complex using ICP-AES. To see the effect of the binding of Ca, the authors measured the spectroscopy and thermostability of the sample with/without the EDTA treatment. The authors have also performed structural analysis of samples obtained by two independent purification methods, and therefore, the reliability of the structures and conclusions drawn is good.

However, it is not sure that the EDTA treatment removed all Ca of the supercomplex, and EDTA may have removed Mg atoms from the supercomplex since EDTA can also work as a chelator for Mg. The author should include ICP-AEC results, at least, of a sample after the EDTA treatment. If possible, ICP-AEC analysis of the LH1-RC sample cultivated in the absence of Ca ions (Ca²⁺-free cells) would be essential to exclude the possibility of Ca²⁺ contamination from reagents.

Minor points:

In Table S2, a comparison of the distances of His to Bchl-Mg is shown. Although these differences are not mentioned in the manuscript, the reviewer believes these differences are considerably influenced by the constraints applied in the structural refinement. Notably, these differences are likely to arise from the differences in the refinement procedure rather than indicative of differences in the structure among phototropic bacteria.

Response to reviewers:

Reviewer #1

Reviewer #1's comments:

Tani et al. describe a structural and biochemical characterization of the LHI-RC complex from *Allochromatium vinosum*. The complex contains various unexpected Ca ions bound to certain alpha and beta subunits of LHI, which suggests that Ca ion binding sites may be a common feature in Chromatiaceae. The role of these Ca ions are determined which especially influence the thermostability of the LHI-RC complexes.

In general, the structure is quite similar to that found in *Allochromatium tepidum*, which slightly diminishes the significance. However, the approaches to determine the roles of the Ca ions, the interpretation of the results, and the communication of that information in the context of existing literature are excellent. The manuscript therefore helps the field to better understand these well studied complexes in a new light.

Especially noteworthy are the following:

- (a) the high resolution achieved that allows for reliable structural determination (e.g., differentiating LHI alpha and beta subunits) and subsequent comparison with other structures,
- (b) the use of ICP-AES to determine the identity of metal ions in the structure, and
- (c) the comparison and biochemical characterization of the complexes with and without calcium

Thus, I enthusiastically recommend the manuscript for publication. My only concern is the use of terms like “best-studied” and “best-understood”, especially in the title. These are somewhat subjective, so I suggest removing them.

Our response:

We appreciate reviewer #1's strong support of our paper. To address the reviewer's concern, we have changed the title and relevant words in the revised manuscript.

Reviewer #2

Reviewer #2's comments: Major point

This manuscript is exciting as it reports the cryo-EM structure of LH1-RC isolated from the mesophilic purple sulfur photosynthetic bacterium *Alc. Vinosum*. This bacterium has been actively studied as a model organism for non-oxygenic photosynthesis, but the resolution of the structure of LH1 has been limited to a 10 Å resolution. In this paper, the authors report a high-resolution cryo-EM structure of LH1-RC isolated from *Alc. Vinosum*. The findings by Tani et al. fill this knowledge gap and will be helpful to photosynthesis researchers. The authors found that the overall structure of *Alc. Vinosum* was very similar to that of *Alc. Tepidum*, but there were differences in the Ca binding site and mode of binding. The authors investigated the relationship between Ca and thermal stability by biochemical and spectroscopic experiments. The results indicate that Ca binding is a common factor for the LH1 complex to confer thermal stability to Chromatiaceae species. This may be of interest to structural biologists in general, as it may be one of the strategies by which protein complexes acquire thermal stability and adapt to their environment. Considering the specialized topics of the journal

Communications Biology and its broad readers, this paper is worthy of publication. However, the reviewer feels that the following points should be addressed before accepting the manuscript.

The LH1-RC supercomplex from *Alc. vinosum* contains multiple polypeptides of $\alpha 1$, $\alpha 2$, $\alpha 3$ and polypeptides of $\beta 1$ and $\beta 2$. The composition of these polypeptides in the supercomplex and the basis of supercomplex formation are revealed in this manuscript. Namely, $\alpha 1$ - binds specifically to $\beta 3$, and $\alpha 2$ specifically to $\beta 1$, which is also seen in the *Alc. tepidum* LH1-RC structure. The structure also shows that only $\alpha 1$ binds Ca with $\beta 1$ and $\beta 3$. As a result, six Ca-binding sites are present in the supercomplex. Furthermore, the authors estimated the number of Ca, Mg, and Fe in the complex using ICP-AES. To see the effect of the binding of Ca, the authors measured the spectroscopy and thermostability of the sample with/without the EDTA treatment. The authors have also performed structural analysis of samples obtained by two independent purification methods, and therefore, the reliability of the structures and conclusions drawn is good.

However, it is not sure that the EDTA treatment removed all Ca of the supercomplex, and EDTA may have removed Mg atoms from the supercomplex since EDTA can also work as a chelator for Mg. The author should include ICP-AEC results, at least, of a sample after the EDTA treatment. If possible, ICP-AEC analysis of the LH1-RC sample cultivated in the absence of Ca ions (Ca²⁺-free cells) would be essential to exclude the possibility of Ca²⁺ contamination from reagents.

Our response:

- We appreciate the reviewer's positive assessment and support of our work. The effects of EDTA on Mg are negligible because (i) the EDTA-treated LH1-RC can be completely converted to its native form upon addition of Ca²⁺ (Fig. 4a, Supplementary Fig. 1a), and (ii) using EGTA, a chelator that has stability constants of $Mg^{2+} \ll Ca^{2+}$, yielded the same result.
- To address the reviewer's concern, we have conducted an additional ICP-AES measurement using EDTA on the LH1-RC from *Alc. vinosum* cultivated in the presence of Ca²⁺. The purified LH1-RC was treated with 20 mM EDTA for 1 hour on ice followed by 10-fold dilution/concentration for three times to remove Ca-EDTA in the sample solution. The result revealed that the Ca²⁺ content reduced to one-fourth of that in the untreated sample, corresponding to 1~2 Ca²⁺ per LH1-RC. However, we are still not sure whether these detected Ca ions represent the tightly bound ones on LH1-RC or contained a part of the residual Ca-EDTA in the sample solution and/or the Ca²⁺ contamination from reagents as pointed out by the reviewer because the total Ca content was close to the noise level in the ICP-AES experiment. Since the result is inconclusive, we prefer not to include it in the manuscript.
- We also did try to analyze Ca contents in the LH1-RC from *Alc. vinosum* cultivated in the absence of Ca²⁺ (9th subculture), but failed to purify enough amounts of normal LH1-RC for ICP-AES measurement, although the cells exhibited a normal absorption spectrum. Phospholipid analysis of the chromatophore membranes from Ca²⁺-free cells revealed that its phospholipid composition had significantly changed (unpublished data), and this may explain the difficulties we encountered in solubilizing the LH1-RC complex in a healthy form.

Reviewer #2's comments: Minor point

In Table S2, a comparison of the distances of His to Bchl-Mg is shown. Although these differences are not mentioned in the manuscript, the reviewer believes these differences are considerably influenced by the constraints applied in the structural refinement. Notably, these differences are likely to arise from the differences in the refinement procedure rather than indicative of differences in the structure among phototropic bacteria.

Our response:

From the published structures of LH1 complexes, we have found a correlation between the LH1 (BChl) Mg–Mg distances and LH1 Q_y absorption maxima (Ref. 30). That is, differences between the average intra- and inter-Mg–Mg distances larger than 0.8 Å tend to yield an LH1 Q_y in the 870–890 nm range, whereas differences smaller than 0.8 Å (more homogeneous Mg–Mg distances) typically yield an LH1 Q_y beyond 900 nm for both BChl *a*- and BChl *b*-containing LH1 complexes. Although this is also the case for *Alc. vinosum* LH1 (we have added a description on this point in the revised manuscript on page 9–10), we are curious whether such correlation applies to the LH1 His–Mg distances and LH1 Q_y transition; i.e., whether more homogeneous (or longer) coordination lengths between α- and β-polypeptides would result in a more redshifted LH1 Q_y. Because the *Alc. vinosum* LH1–RC structure was solved at high resolution (2.24 Å), we believe that the LH1 His–Mg distances listed in our paper would be less affected by the refinement procedure and parameters used, thus, clarifying this issue. However, as pointed out by the reviewer, since the published structures of LH1 were solved at various resolutions using presumably different parameters between researcher groups, it is difficult to evaluate the effects at present. For this reason, we have made efforts to obtain high-resolution structural data including the His–Mg distances toward a comprehensive understanding of LH1 structure/function relationships.

REVIEWERS' COMMENTS:

Reviewer #2 (Remarks to the Author):

The authors have satisfactorily addressed all concerns and revised manuscript accordingly. Therefore, this manuscript meets the standards of Communication Biology.